# Screening Potato Landraces to Cope with Climate Change in the Central Andes of Peru

Edith Zevallos [1], Karina Marmolejo [1], Fernando Alvarez [1], Rocio Paitan [1], Ines Viza [1], Dante Becerra [1], Gina Rixi [1], Cecilia Silva-Diaz [2] and Josue Inga[1,*]

[1] Departamento de Agronomía, Facultad de Ciencias Agropecuarias, Universidad Nacional Daniel Alcides Carrión (UNDAC), San Juan de Jarampa s/n, Pasco 19000, Peru; ezevallosa@undac.edu.pe (E.Z.); kmarmolejog@undac.edu.pe (K.M.); falvarezr@undac.edu.pe (F.A.); rpaitang@undac.edu.pe (R.P.); ivizap@undac.edu.pe (I.V.); dbecerrap@undac.edu.pe (D.B.); grixiv@undac.edu.pe (G.R.)

[2] Agricultural Sciences and Resource Management in the Tropics and Subtropics (ARTS), University of Bonn, 53113 Bonn, Germany; silvadiaz.cc@gmail.com

* Correspondence: jingaortiz@undac.edu.pe; Tel.: +51-971-231-179

**Abstract:** Agriculture in the Andean region is mainly small-scale and rainfed, especially in Peru where almost 80% of its population depends on agriculture-related activities. Climate change in addition to social factors threatens the food security of this region. The forecast of more frequent dry spells would especially affect potato crops, domesticated centuries ago in the Andes, where there remains a great genetic diversity. This study aimed to characterize the response to drought stress of 79 potato landraces traditionally grown in the Central Andes of Peru (Pasco region) as a first selection for future breeding evaluations. The used indices were mean productivity (MP), geometric mean productivity (GMP), stress tolerance (STI), stress susceptibility (SSI), and tolerance index (TOL), and a scoring methodology that integrates all of them into a single descriptor in a simple and fast way. The varied responses showed a wide genetic diversity within the assessed landraces, where at least nine of them own high resilience and productivity qualities, and many others are highly vulnerable to drought. It is recommended to complement these studies with physiological and molecular evaluations in stress situations, especially in those with tolerance qualities highlighted in this study, and thus promote the conservation of the biodiversity of this region.

**Keywords:** rainfed agriculture; drought stress; food security; tolerance; resilience; potato yield

## 1. Introduction

Agriculture consumes on average 70% of the world's available freshwater, and in Peru, this figure rises to 85% [1], where almost 80% of its population depends on agriculture-related activities [2]. According to an agricultural census [3], 65% of the Peruvian agricultural area lacks irrigation infrastructure, mainly depending on rainwater. Historically, the Peruvian Andes have been characterized by drought episodes that have reduced agricultural production by up to 70% [4], making its population particularly vulnerable. Climate change could aggravate this situation with much less water available if the glaciers disappear, rainfall is reduced and temperatures rise causing an increase in evaporation [5]. Further, a strengthening of the tropical atmospheric circulation would cause the tropical Andes to become wetter and the subtropical Andes drier [6]. The rainy season in the Peruvian Andes runs from November to April, where precipitation gradually increases between October and November, ending abruptly in April. However, in recent years, the rain onset has not occurred in any of these months, but until December or January (E. Zevallos, personal communication, 7 July 2023). Additionally, the region experiences a degree of continental rising effect and transition to an extra-tropical zone, which is sometimes subject to prolonged cold waves due to the outbreak of polar air masses to the north [7].

Potato is one of the leading temporary crops in Peru that, together with rice, yellow dent corn, starchy corn, and grain barley, covered 57.4% of the total agricultural area planted in 2021 [8]. This crop, whose origin and genetic diversification were developed in the Andes, is still preserved in-situ [9], susceptible to water availability, and threatened by the effects of climate change [10]. Native potato landraces are genotypes adapted and maintained by farmers under heterogeneous conditions in the Andes [9]. Being strongly embedded in the local culture, these varieties constitute the backbone of food security and subsistence in the Andean region [11], especially in Peru, where there are around 6000 communities of small-scale farmers [12].

Climate change is altering the growing conditions of potatoes at an unprecedented speed and intensity through biotic and abiotic stressors, in addition to the impact that Andean communities suffer from environmental degradation, urbanization, and changes in diet preferences and lifestyle [11]. Farmers value a high level of crop diversity for culinary purposes, and there is a need to evaluate at the farmer level the decision-making process that dictates persistence or loss of diversity [13]. The conservation of potato landraces is also at risk due to a generational gap, since the youth of these communities prefer education and working in urban areas, breaking the inter-generational transmission of traditional knowledge about potato production [11]. Therefore, it is crucial to create resilient production systems that ensure farmers' food security [14], in addition to the fact that diversity-based risk mitigation is a realistic way to adapt to environmental change [13]. One way to do this is to recognize potato varieties with drought tolerance qualities, use them in genetic improvement, and make appropriate decisions [15,16]. If stress and non-stress events occur with the same frequency in a region, such as recurrent droughts in the Andes, selection for tolerance would be useful [17].

Some varieties of native potatoes cultivated above 3500 masl, present natural adaptation to adverse factors such as drought; hence, their genes present a high potential in the selection of varieties [18–20]. This has been demonstrated by Cabello et al. [21], who subjected 918 potato accessions (clustered into improved varieties, genetic stocks, and landraces), to a 62% reduction in water supply, resulting in an average yield reduction of 58%, but landraces were only affected by 38%. Later, these accessions were evaluated with the indices mean productivity (MP), geometric mean productivity (GMP), tolerance index (TOL), drought tolerance index (DTI), drought susceptibility index (DSI), and yield stability index (YSI), where MP, GMP, and DTI were the best for identifying genotypes with high yield and tolerance to drought [22]. However, the scientific information about the conservation of potato landraces by small-scale farmers is scarce [11], evidencing a gap between traditional knowledge and science. This study aimed to reduce this gap by describing the response to drought stress of 79 potato landraces traditionally grown in the central Andes of Peru (Pasco region) based on their yield and tolerance indices previously tested in this and other crops.

## 2. Materials and Methods

### 2.1. Location

The experiment was carried out in the greenhouse of the School of Agronomy of the National University Daniel Alcides Carrion (UNDAC), in the district of Paucartambo (10°46′14.1″ S, 75°48′53.4″ W, 2824 masl, Pasco, Peru), from 18 August 2021 to 10 March 2022. The region is characterized by an annual rainfall of 1010.1 mm, and average relative humidity of $60.7 \pm 1.4$%, in addition to average values of the maximum, medium, and minimum temperature of $19.0 \pm 0.2$, $11.8 \pm 0.2$, and $7.0 \pm 2.1$ °C, respectively [23]. Within the greenhouse covered by a clear polyethylene film, the minimum and maximum temperatures recorded during the trial were 6.7 and 39.9 °C, respectively, and the relative humidity ranged from 1 to 99.2%, recorded with a Hakusa Hw521digital thermometer-hygrometer.

### 2.2. Plant Material and Experimental Design

The 79 genotypes evaluated in this experiment were a selection based on performance and site availability from 120 potato landraces of the Pasco region preserved (in situ) by the Native potato-UNDAC project, and identified by folk names assigned by local farmers in Quechua and Spanish languages. Some were also part of a late blight resistance study in the same region [24]. Each experimental unit was planted in a 7-liter pot following a completely randomized block design with three replicates and covering a total area of 200 m$^2$. The substrate (4.5 kg) was composed of agricultural soil and sand in a 2:1 ratio and irrigated at field capacity (2 L) when tensiometers showed 20 cbar, initially once per week, and twice in later stages. Eighty days after sowing—DAS (November 6), when flowering and tuberization, the most sensitive stages to drought stress [25], were estimated to be close, irrigation was suspended for 16 days, restricting 4 consecutive irrigation pulses, to simulate drought stress conditions (45 cbar) [26] in the treatment pots, and leaving the pots irrigated at field capacity as controls. At 212 DAS, the harvest was carried out to assess fresh tuber yield per plant.

### 2.3. Evaluation of Tolerance to Drought Stress

The yield values based on fresh tuber weight per plant (g plant$^{-1}$) with field capacity (control) and restricted irrigation (treatment) were used to calculate the following indices [17,27–29]:

$$\text{Arithmetic mean productivity (MP)} = (Yc + Yr)/2 \tag{1}$$

$$\text{Geometric mean productivity (GMP)} = (Yc \times Yr)^{0.5} \tag{2}$$

$$\text{Tolerance index (TOL)} = Yc - Yr \tag{3}$$

$$\text{Stress tolerance index (STI)} = (Yr \times Yc)/Xc^2 \tag{4}$$

$$\text{Stress susceptibility index (SSI)} = (1 - Yr/Yc)/(1 - Xr/Xc) \tag{5}$$

where $Yc$ and $Yr$ correspond to the yield of a genotype with field capacity and restricted irrigation, respectively. $Xc$ and $Xr$ correspond to the average yield of all the genotypes under field capacity and restricted irrigation, respectively.

Later, these indices were scored following the criteria of [30] in order to combine the information provided by each of them and give a solid characterization of the genotypes. The scores range from 1 to 10, adjusting the scale with the minimum and maximum values obtained for each index within the total population studied. To establish that high scores meant a "good genotype", the TOL and SSI values were inverted due to their negative relationship with yield. To work with large germplasm sets as in this study, an R function (available in [31]) was used. This function also calculates the 18 possible combinations of scored indices proposed by Thiry et al. [30], and indicates the best correlated with the original variables (yield under stress and non-stress conditions) that would reflect better the resilience (drought tolerance) and productivity (higher yield) capacity of each genotype.

### 2.4. Statistical Analysis

The fresh tuber yield under restricted irrigation was subjected to an analysis of variance, followed by a Tukey test (HSD), to identify which genotypes differed significantly from the others. These analyses were carried out with the "Agricolae" R package [32]. The relationship between the calculated indices and the yield under restricted irrigation was analyzed with a correlation analysis. Likewise, the correlation between these indices was analyzed.

## 3. Results

### 3.1. The Yield of Potatoes Landraces in Different Scenarios

The average yield of the 79 potato landraces irrigated at field capacity ($Yc$-control) was 303 g plant$^{-1}$, with maximum and minimum values of 523 and 134 g plant$^{-1}$, respectively. Among them, only three genotypes exceeded 500 g plant$^{-1}$ (Cantiña, Sumaq sunqu rojo, and Leona). The yield of the complete set of genotypes under restricted irrigation ($Yr$) ranged between 20 and 282 g plant$^{-1}$ with significant differences within them (F = 2.54), especially for the top five with a $Yr$ close to 50% Yc. The highest values corresponded to Sumaq sunqu rojo, Huayro negro, Yana galla shaco, Huayro plomo, and Viuda, which yielded above 245 g plant$^{-1}$ (Table 1). As for the reduction in yield due to restricted irrigation, half of the landraces reduced their yield (control) by more than 50%, and in the worst case, they exceeded 80% reduction, although there were those that only reduced their yield by less than 20% (Figure 1).

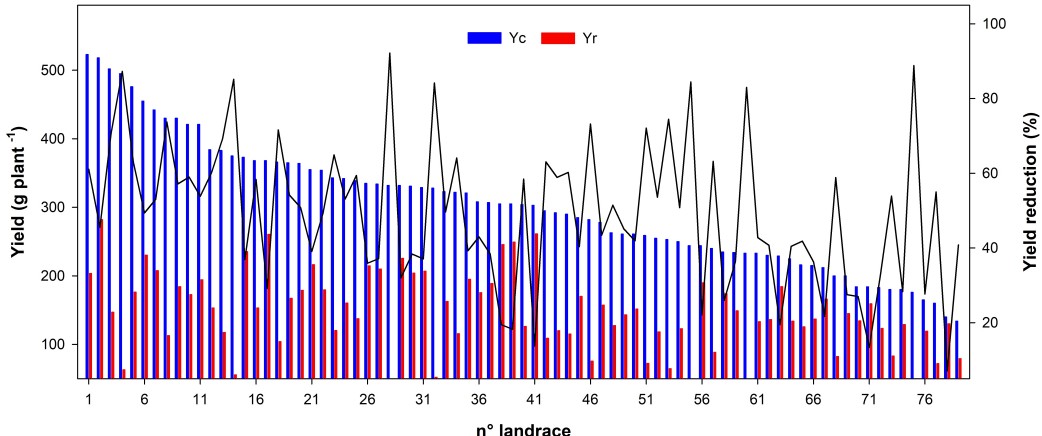

**Figure 1.** Yield irrigated at field capacity ($Yc$, control) and with restricted irrigation ($Yr$, treatment) of the 79 potato landraces traditionally grown in Pasco, Peru. The percentage reduction in yield due to induced stress is also indicated with the black line. The numbers assigned to each landrace are indicated in Table 1.

### 3.2. Tolerance Indices Based on Yield

The responses of potato landraces in both scenarios (stress and non-stress conditions) were integrated into the aforementioned indices, and which absolute values for each landrace are presented in Table 1. MP, GMP, and STI were highly positively correlated with yield under stress ($Yr$), and SSI showed a negative relationship with $Yr$ (Figure 2). TOL was only highly correlated with fully irrigated yield ($Yc$). Additionally, the correlation analysis between the indices showed a highly significant relationship between STI and GMP (0.98), GMP and MP (0.95), and SSI and TOL (0.83).

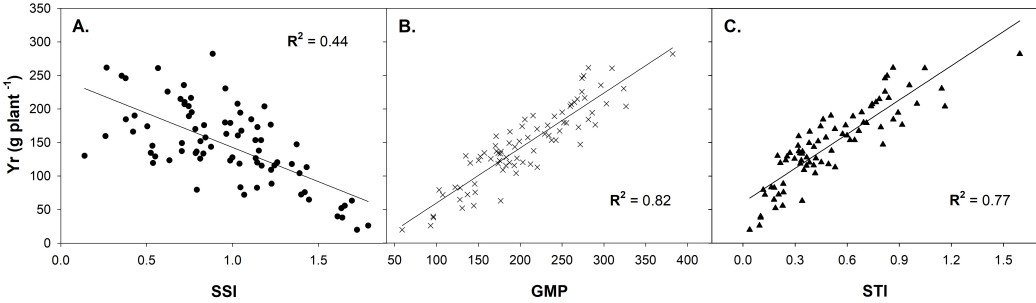

**Figure 2.** Relationship between yield with restricted irrigation ($Yr$) and (**A**) stress susceptibility index (SSI), (**B**) geometric mean productivity (GMP), and (**C**) stress tolerance index (STI) in 79 potato landraces traditionally grown in Pasco, Peru.

**Table 1.** Indices of tolerance to drought stress in 79 potato landraces traditionally grown in Pasco, Peru.

| n° | Landraces | *Yc* | *Yr* | HSD | MP | GMP | TOL | STI | SSI |
|----|-----------|------|------|-----|-----|-----|-----|-----|-----|
| 2 | Sumaq sunqu rojo | 518 | 282.0 | a | 400.0 | 382.2 | 236.0 | 1.6 | 0.9 |
| 41 | Huayro negro | 303 | 261.3 | ab | 282.2 | 281.4 | 41.7 | 0.9 | 0.3 |
| 16 | Yana galla shaco | 368 | 260.7 | abc | 314.3 | 309.7 | 107.3 | 1.0 | 0.6 |
| 38 | Huayro plomo | 305 | 249 | abcd | 277.2 | 275.8 | 55.7 | 0.8 | 0.4 |
| 39 | Viuda | 305 | 246 | abcde | 275.3 | 273.7 | 59.3 | 0.8 | 0.4 |
| 15 | Shogo chata | 373 | 235.3 | abcdef | 304.2 | 296.3 | 137.7 | 1.0 | 0.7 |
| 6 | Gara callhuan | 455 | 230.3 | abcdef | 342.7 | 323.7 | 224.7 | 1.1 | 1.0 |
| 28 | Uncuy | 332 | 225.7 | abcdef | 278.8 | 273.7 | 106.3 | 0.8 | 0.6 |
| 21 | Muru dólar | 355 | 216.3 | abcdef | 285.7 | 277.1 | 138.7 | 0.8 | 0.8 |
| 26 | Niña papa | 335 | 214.7 | abcdef | 274.8 | 268.2 | 120.3 | 0.8 | 0.7 |
| 27 | Puka cauriña | 334 | 210.0 | abcdef | 272.0 | 264.8 | 124.0 | 0.8 | 0.7 |
| 7 | Azul callhuan | 442 | 207.7 | abcdef | 324.8 | 303.0 | 234.3 | 1.0 | 1.0 |
| 31 | Azulino | 329 | 207.0 | abcdef | 268.0 | 261.0 | 122.0 | 0.7 | 0.7 |
| 30 | Muru ranra ñahui | 331 | 204.0 | abcdef | 267.5 | 259.9 | 127.0 | 0.7 | 0.7 |
| 1 | Cantiña | 523 | 203.7 | abcdef | 363.3 | 326.4 | 319.3 | 1.2 | 1.2 |
| 35 | Yana shenga | 321 | 195.0 | abcdef | 258.0 | 250.2 | 126.0 | 0.7 | 0.8 |
| 10 | Rayhuana | 421 | 194.3 | abcdef | 307.7 | 286.0 | 226.7 | 0.9 | 1.0 |
| 55 | Galleta blanca | 244 | 190.0 | abcdef | 217.0 | 215.3 | 54.0 | 0.5 | 0.4 |
| 37 | Muru pillush | 307 | 189.0 | abcdef | 248.0 | 240.9 | 118.0 | 0.6 | 0.7 |
| 8 | Muru piña | 430 | 184.3 | abcdef | 307.2 | 281.5 | 245.7 | 0.9 | 1.1 |
| 63 | Cacho de toro | 229 | 184.3 | abcdef | 206.7 | 205.5 | 44.7 | 0.5 | 0.4 |
| 22 | Morales rojo | 354 | 179.7 | abcdef | 266.8 | 252.2 | 174.3 | 0.7 | 1.0 |
| 20 | Puka canasta | 364 | 179.0 | abcdef | 271.5 | 255.3 | 185.0 | 0.7 | 1.0 |
| 5 | Huasho | 476 | 176.3 | abcdef | 326.2 | 289.7 | 299.7 | 0.9 | 1.2 |
| 36 | Runtush | 308 | 175.7 | abcdef | 241.8 | 232.6 | 132.3 | 0.6 | 0.8 |
| 58 | Matucana | 235 | 174.0 | abcdef | 204.5 | 202.2 | 61.0 | 0.4 | 0.5 |
| 11 | Acacluy pecho | 421 | 172.7 | abcdef | 296.8 | 269.6 | 248.3 | 0.8 | 1.1 |
| 45 | Tarma | 285 | 170.0 | abcdef | 227.5 | 220.1 | 115.0 | 0.5 | 0.8 |
| 19 | Yuraj morales | 365 | 167.3 | abcdef | 266.2 | 247.1 | 197.7 | 0.7 | 1.1 |
| 67 | Cahuashina moro | 212 | 166.0 | abcdef | 189.0 | 187.6 | 46.0 | 0.4 | 0.4 |
| 33 | Higos | 323 | 162.7 | abcdef | 242.8 | 229.2 | 160.3 | 0.6 | 1.0 |
| 24 | Yuca suytu | 342 | 160.3 | abcdef | 251.2 | 234.2 | 181.7 | 0.6 | 1.0 |
| 70 | Muru puñete | 184 | 159.3 | abcdef | 171.7 | 171.2 | 24.7 | 0.3 | 0.3 |
| 47 | Huayti chuco | 278 | 157.3 | abcdef | 217.7 | 209.1 | 120.7 | 0.5 | 0.8 |
| 12 | Niño suytu | 384 | 153.3 | abcdef | 268.7 | 242.7 | 230.7 | 0.6 | 1.2 |
| 17 | Yuraj pillish | 368 | 153.3 | abcdef | 260.7 | 237.5 | 214.7 | 0.6 | 1.1 |
| 49 | Alcarraz | 261 | 151.7 | abcdef | 206.3 | 199.0 | 109.3 | 0.4 | 0.8 |
| 59 | Cahuashina | 234 | 149.0 | abcdef | 191.5 | 186.7 | 85.0 | 0.4 | 0.7 |
| 3 | Leona | 502 | 147.0 | abcdef | 324.5 | 271.7 | 355.0 | 0.8 | 1.4 |
| 68 | Yawar taico | 200 | 145.0 | abcdef | 172.5 | 170.3 | 55.0 | 0.3 | 0.5 |
| 50 | Piña negra | 261 | 143.3 | abcdef | 202.2 | 193.4 | 117.7 | 0.4 | 0.9 |
| 25 | Chaucha | 339 | 137.7 | abcdef | 238.3 | 216.0 | 201.3 | 0.5 | 1.2 |
| 66 | Huayro rojo | 215 | 137.0 | abcdef | 176.0 | 171.6 | 78.0 | 0.3 | 0.7 |
| 62 | Milagro rojo | 230 | 136.3 | abcdef | 183.2 | 177.1 | 93.7 | 0.3 | 0.8 |
| 71 | Puka pampiña | 184 | 134.3 | abcdef | 159.2 | 157.2 | 49.7 | 0.3 | 0.5 |
| 64 | Yana semita | 225 | 134.0 | abcdef | 179.5 | 173.6 | 91.0 | 0.3 | 0.8 |
| 60 | Puka dólar | 233 | 133.3 | abcdef | 183.2 | 176.3 | 99.7 | 0.3 | 0.8 |
| 78 | Jilguero rojo | 140 | 130.0 | abcdef | 135.0 | 134.9 | 10.0 | 0.2 | 0.1 |
| 73 | Muru tarma | 180 | 129.0 | abcdef | 154.5 | 152.4 | 51.0 | 0.3 | 0.6 |
| 48 | Santo domingo | 263 | 127.7 | abcdef | 195.3 | 183.2 | 135.3 | 0.4 | 1.0 |
| 40 | Chaulina | 304 | 126.3 | abcdef | 215.2 | 196.0 | 177.7 | 0.4 | 1.1 |
| 65 | Clavel suytu | 216 | 125.7 | abcdef | 170.8 | 164.8 | 90.3 | 0.3 | 0.8 |
| 72 | Huayro blanco | 183 | 123.3 | abcdef | 153.2 | 150.2 | 59.7 | 0.2 | 0.6 |
| 54 | Huayro moro rojo | 250 | 123.0 | abcdef | 186.5 | 175.4 | 127.0 | 0.3 | 1.0 |
| 23 | Yana pillush | 343 | 120.3 | abcdef | 231.7 | 203.2 | 222.7 | 0.5 | 1.3 |
| 43 | Yana huaca | 292 | 120.0 | abcdef | 206.0 | 187.2 | 172.0 | 0.4 | 1.1 |
| 76 | Peruanita | 165 | 119.3 | abcdef | 142.2 | 140.3 | 45.7 | 0.2 | 0.5 |

**Table 1.** *Cont.*

| n° | Landraces | Yc | Yr | HSD | MP | GMP | TOL | STI | SSI |
|---|---|---|---|---|---|---|---|---|---|
| 52 | Huanuqueña | 255 | 118.3 | abcdef | 186.7 | 173.7 | 136.7 | 0.3 | 1.0 |
| 13 | Chiaquil rojo | 383 | 117.7 | abcdef | 250.3 | 212.3 | 265.3 | 0.5 | 1.3 |
| 34 | Camotillo | 322 | 115.7 | abcdef | 218.8 | 193.0 | 206.3 | 0.4 | 1.2 |
| 44 | Cauriña | 290 | 115.3 | abcdef | 202.7 | 182.9 | 174.7 | 0.4 | 1.2 |
| 9 | Cuchipa ismaynan | 430 | 113.0 | abcdef | 271.5 | 220.4 | 317.0 | 0.5 | 1.4 |
| 42 | Jilguero | 295 | 109.0 | abcdef | 202.0 | 179.3 | 186.0 | 0.4 | 1.2 |
| 18 | Piña morada | 366 | 104.0 | abcdef | 235.0 | 195.1 | 262.0 | 0.4 | 1.4 |
| 57 | Gargash suytu | 240 | 88.3 | abcdef | 164.2 | 145.6 | 151.7 | 0.2 | 1.2 |
| 74 | Shashai warmi | 180 | 83.0 | abcdef | 131.5 | 122.2 | 97.0 | 0.2 | 1.0 |
| 69 | Yana cauriña | 200 | 82.3 | abcdef | 141.2 | 128.3 | 117.7 | 0.2 | 1.1 |
| 79 | Añaspi yawarmi | 134 | 79.3 | abcdef | 106.7 | 103.1 | 54.7 | 0.1 | 0.8 |
| 46 | Llama chupan | 282 | 75.7 | abcdef | 178.8 | 146.1 | 206.3 | 0.2 | 1.4 |
| 51 | Piña blanca | 259 | 72.3 | abcdef | 165.7 | 136.9 | 186.7 | 0.2 | 1.4 |
| 77 | Sunic | 160 | 72.0 | abcdef | 116.0 | 107.3 | 88.0 | 0.1 | 1.1 |
| 53 | Galleta | 253 | 64.7 | abcdef | 158.8 | 127.9 | 188.3 | 0.2 | 1.4 |
| 4 | Shiri | 495 | 63.0 | abcdef | 279.0 | 176.6 | 432.0 | 0.3 | 1.7 |
| 14 | Chiaquil negro | 375 | 55.7 | bcdef | 215.3 | 144.5 | 319.3 | 0.2 | 1.7 |
| 32 | Orgon runtush | 328 | 52.0 | bcdef | 190.0 | 130.6 | 276.0 | 0.2 | 1.6 |
| 61 | Merino | 233 | 39.7 | cdef | 136.3 | 96.1 | 193.3 | 0.1 | 1.6 |
| 56 | Puka ranra ñahui | 244 | 38.0 | def | 141. | 96.3 | 206.0 | 0.1 | 1.6 |
| 29 | Huanuco suytu | 332 | 26.0 | ef | 179.0 | 92.9 | 306.0 | 0.1 | 1.8 |
| 75 | Cera monilla | 176 | 19.7 | f | 97.8 | 58.8 | 156.3 | 0.0 | 1.7 |

n°: order number regarding the yield irrigated at field capacity, also in Figure 1. *Yc*: yield irrigated at field capacity (control). *Yr*: yield with restricted irrigation. HSD: significant difference according to a Tukey's test based on *Yr*. MP: mean productivity index. GMP: geometric mean productivity index. TOL: tolerance index. STI: stress tolerance index. SSI: stress susceptibility index.

### 3.3. Scoring the Indices Responses

To better visualize and compare the information provided by each index, whose ranges of values were quite different, a scoring was applied. The newly scored indices were also correlated with their corresponding originals, which showed a high correlation (0.99), this being negative for SSI and TOL and positive for the rest. The "good genotypes" with the highest rankings (green color), or better responses were Huayro negro, Sumaq sunqu rojo, and Yana galla shaco. On the opposite side of the same table, the lowest rankings (red color) were for Puka ranra ñahui and Cera monilla (Table 2). Finally, the best combination ($R^2 = 0.92$) of the scored indices was represented by the Yield Stress Score Index (YSSI), which makes an easy differentiation between the assessed genotypes, obtained as follows:

$$(SSI + MP)/2 = YSSI \tag{6}$$

**Table 2.** Scored (S_) tolerance indices (detailed in Table 1) based on the responses to drought stress of 79 potato landraces traditionally grown in Pasco, Peru. Green, yellow and red tones indicate the best, middle and worst qualities, respectively. YSSI (Yield Stress Score Index) integrates the responses of the scored indices.

| Landraces | S_SSI | S_TOL | S_MP | S_GMP | S_STI | YSSI |
|---|---|---|---|---|---|---|
| Huayro negro | 10 | 10 | 7 | 7 | 6 | 8.5 |
| Sumaq sunqu rojo | 6 | 5 | 10 | 10 | 10 | 8 |
| Yana galla shaco | 8 | 8 | 8 | 8 | 7 | 8 |
| Viuda | 9 | 9 | 6 | 7 | 6 | 7.5 |
| Gara callhuan | 6 | 5 | 9 | 9 | 8 | 7.5 |
| Huayro plomo | 9 | 9 | 6 | 7 | 6 | 7.5 |
| Shogo chata | 7 | 7 | 7 | 8 | 6 | 7 |
| Uncuy | 8 | 8 | 6 | 7 | 6 | 7 |
| Muru dólar | 7 | 7 | 7 | 7 | 6 | 7 |

**Table 2.** *Cont.*

| Landraces | S_SSI | S_TOL | S_MP | S_GMP | S_STI | YSSI |
|---|---|---|---|---|---|---|
| Cantiña | 4 | 3 | 9 | 9 | 8 | 6.5 |
| Azulino | 7 | 8 | 6 | 7 | 5 | 6.5 |
| Muru ranra ñahui | 7 | 8 | 6 | 7 | 5 | 6.5 |
| Niña papa | 7 | 8 | 6 | 7 | 5 | 6.5 |
| Muru puñete | 10 | 10 | 3 | 4 | 2 | 6.5 |
| Cacho de toro | 9 | 10 | 4 | 5 | 3 | 6.5 |
| Puka cauriña | 7 | 8 | 6 | 7 | 5 | 6.5 |
| Galleta blanca | 9 | 9 | 4 | 5 | 4 | 6.5 |
| Cahuashina moro | 9 | 10 | 4 | 4 | 3 | 6.5 |
| Yana shenga | 7 | 8 | 6 | 6 | 5 | 6.5 |
| Azul callhuan | 5 | 5 | 8 | 8 | 7 | 6.5 |
| Tarma | 7 | 8 | 5 | 5 | 4 | 6 |
| Morales rojo | 6 | 7 | 6 | 6 | 5 | 6 |
| Huasho | 4 | 4 | 8 | 8 | 6 | 5 |
| Jilguero rojo | 10 | 10 | 2 | 3 | 2 | 6 |
| Matucana | 8 | 9 | 4 | 5 | 3 | 6 |
| Muru piña | 5 | 5 | 7 | 7 | 6 | 6 |
| Rayhuana | 5 | 5 | 7 | 8 | 6 | 6 |
| Muru pillush | 7 | 8 | 5 | 6 | 4 | 6 |
| Leona | 3 | 2 | 8 | 7 | 5 | 5.5 |
| Cahuashina | 7 | 9 | 4 | 4 | 3 | 5.5 |
| Yuca suytu | 5 | 6 | 6 | 6 | 4 | 5.5 |
| Higos | 6 | 7 | 5 | 6 | 4 | 5.5 |
| Acacluy pecho | 4 | 5 | 7 | 7 | 5 | 5.5 |
| Yuraj morales | 5 | 6 | 6 | 6 | 5 | 5.5 |
| Yawar taico | 8 | 9 | 3 | 4 | 2 | 5.5 |
| Runtush | 6 | 8 | 5 | 6 | 4 | 5.5 |
| Puka pampiña | 8 | 10 | 3 | 4 | 2 | 5.5 |
| Puka canasta | 5 | 6 | 6 | 7 | 5 | 5.5 |
| Alcarraz | 6 | 8 | 4 | 5 | 3 | 5 |
| Muru tarma | 8 | 10 | 2 | 3 | 2 | 5 |
| Peruanita | 8 | 10 | 2 | 3 | 2 | 5 |
| Yana semita | 7 | 9 | 3 | 4 | 2 | 5 |
| Milagro rojo | 7 | 9 | 3 | 4 | 2 | 5 |
| Niño suytu | 4 | 5 | 6 | 6 | 4 | 5 |
| Piña negra | 6 | 8 | 4 | 5 | 3 | 5 |
| Huayti chuco | 6 | 8 | 4 | 5 | 3 | 5 |
| Huayro rojo | 7 | 9 | 3 | 4 | 2 | 5 |
| Huayro blanco | 8 | 9 | 2 | 3 | 2 | 5 |
| Yuraj pillish | 7 | 9 | 3 | 4 | 2 | 5 |
| Chiaquil rojo | 3 | 4 | 6 | 5 | 3 | 4.5 |
| Santo domingo | 5 | 8 | 4 | 4 | 3 | 4.5 |
| Chaucha | 4 | 6 | 5 | 5 | 4 | 4.5 |
| Camotillo | 4 | 6 | 5 | 5 | 3 | 4.5 |
| Clavel suytu | 6 | 9 | 3 | 4 | 2 | 4.5 |
| Puka dólar | 6 | 8 | 3 | 4 | 2 | 4.5 |
| Cuchipa ismaynan | 3 | 3 | 6 | 5 | 4 | 4.5 |
| Yana pillush | 4 | 5 | 5 | 5 | 3 | 4.5 |
| Huanuqueña | 5 | 7 | 3 | 4 | 2 | 4 |
| Yana huaca | 4 | 7 | 4 | 4 | 3 | 4 |
| Chaulina | 4 | 7 | 4 | 5 | 3 | 4 |
| Añaspi yawarmi | 7 | 9 | 1 | 2 | 1 | 4 |
| Piña morada | 3 | 5 | 5 | 5 | 3 | 4 |
| Huayro moro rojo | 5 | 8 | 3 | 4 | 2 | 4 |
| Cauriña | 4 | 7 | 4 | 4 | 3 | 4 |
| Jilguero | 4 | 6 | 4 | 4 | 3 | 4 |

**Table 2.** *Cont.*

| Landraces | S_SSI | S_TOL | S_MP | S_GMP | S_STI | YSSI |
|---|---|---|---|---|---|---|
| Shiri | 1 | 1 | 6 | 4 | 2 | 3.5 |
| Shashai warmi | 5 | 8 | 2 | 2 | 1 | 3.5 |
| Gargash suytu | 4 | 7 | 3 | 3 | 2 | 3.5 |
| Piña blanca | 3 | 6 | 3 | 3 | 2 | 3 |
| Yana cauriña | 4 | 8 | 2 | 3 | 1 | 3 |
| Galleta | 3 | 6 | 3 | 3 | 1 | 3 |
| Sunic | 5 | 9 | 1 | 2 | 1 | 3 |
| Lama chupan | 3 | 6 | 3 | 3 | 2 | 3 |
| Chiaquil negro | 1 | 3 | 4 | 3 | 2 | 2.5 |
| Orgon runtush | 1 | 4 | 4 | 3 | 1 | 2.5 |
| Merino | 2 | 6 | 2 | 2 | 1 | 2 |
| Huanuco suytu | 1 | 3 | 3 | 2 | 1 | 2 |
| Puka ranra ñahui | 1 | 6 | 2 | 2 | 1 | 1.5 |
| Cera monilla | 1 | 7 | 1 | 1 | 1 | 1 |

Green and red colors indicate the best and worst qualities, respectively, for the study objectives, and yellow tones correspond to those in between.

## 4. Discussion

### 4.1. Yield Gap in the Central Andean Region of Peru

Although these landraces are widely consumed in the central Andes of Peru, many of them in the Junin and Ayacucho regions [33], no studies focused on their molecular or physiological characteristics were found to contrast our results. Yield was the main trait assessed in this study since it represents the outcome of all adaptive mechanisms with which plants respond to stress [34]. In Peru, native potatoes can yield between 500 and 2450 [35] and even exceed 3000 g plant$^{-1}$ (considering 20% of dry matter, [36]) under potential conditions. Ten landraces from this study were also part of another trial that evaluated their response to *Phytophthora infestans* infection in the same region [24], and both stress treatments had similar responses, possibly because the effect of stress on yield is independent of its biotic or abiotic origin [37]. However, the yield irrigated at field capacity in this study ($Yc$) was lower than in previous reports, evidencing the yield gap in the Andean region [38]. Among the causes could be the limited space for optimal tuber development inside the pots [39], insufficient fertilization [40], and low availability of high-quality seeds. Still, the comparison made within a population of 79 landraces would be useful to distinguish those with better and worse qualities for a first selection because they were under the same conditions and time period [30]. In addition, the restricted irrigation pulses were sufficient to distinguish both scenarios, by inducing moderate drought stress that is more recommended than a severe one to detect tolerant genotypes [41–43].

### 4.2. Usefulness of Tolerance Indices

Widely used indices to characterize and select stress-tolerant genotypes consider crop responses under stress and non-stress conditions since a stress-free environment allows for better expression of genetic potential with a high heritability of yield [44]. A good index should identify genotypes capable of reaching good yields under both conditions [45], because a high yield in the absence of stress does not automatically indicate good performance under stress, and a high yield under stress does not precisely indicate high resilience [30]. Genotypes can be classified into four groups according to their responses: (A) uniform and superior in both stress and non-stress conditions, (B) favorable only in the absence of stress, (C) better only in stress conditions, and (D) with poor performance in both conditions [27]. Apparently, group C would meet the selection objectives, but this positive response may not occur under "normal" conditions [27]. Therefore, optimal selection would identify group A [45], as it is more important for subsistence agriculture to have stable yields under different conditions than higher yields only under favorable conditions [17].

The MP, GMP, and STI indices highlighted the same landraces (Table 2) whose average yield exceeded the rest under both conditions. MP favors varieties with high potential

yield strengthened by a stress tolerance [17], which has been verified in chickpeas, wheat, and sorghum [46–48]. However, it is an arithmetic variable that would have an upward bias due to a relatively larger difference between yields irrigated at field capacity ($Yc$) and restricted irrigation ($Yr$). The geometric variable (GMP) is less sensitive to large extreme values, and thus, less dependent on high yields under potential conditions, and would be a better indicator. In this study, MP and GMP were highly correlated and gave the same results, similar to Cabello et al. [22] and Sandaña et al. [45]. However, for Fernandez [27] these indicators would not distinguish between groups A and B.

The TOL index considers the most tolerant those with minimal differences between $Yc$ and $Yr$. Within them is included the landrace Peruanita which has been previously considered tolerant to short periods of drought [49]. But, when focusing on $Yc$, the top list landraces based on TOL, with the exception of Huayro negro, had an average value of 180 g plant$^{-1}$ with no significant differences, well below the highest yields in this study. Fernandez [27] considers that this index cannot distinguish between groups A and C because is not possible to define if the small difference was due to a high yield under limited conditions or a very low yield under optimal conditions. The SSI index considers stress-tolerant genotypes with values less than 1 [50], so more than half in this study (Table 1) would be included within this group. Those on the top had the lowest yield reduction under stress conditions. This index favors genotypes with low potential yield [29] but fails again to distinguish groups A and C [27], similar to TOL, with which it was highly correlated. Still, Cabello et al. [22] considered it important because it not only depends on $Yc$ and $Yr$ but also on the stress intensity that is, on the environmental conditions.

### 4.3. Highlights within Local Diversity

The similar responses and high correlation between some indices support the idea of Thiry et al. [30] to classify them into two classes, the susceptibility indices (TOL and SSI) that distinguish between stress tolerant and susceptible genotypes, and have a negative relationship with yield (Figure 2A), and the tolerance indices (MP, GMP, and STI) that identify genotypes with tolerance and high average yield, having a positive relationship with yield (Figure 2B,C). These last ones, the most recommended to identify genotypes in stress and non-stress environments, especially STI are considered the most appropriate for distinguishing group A from B and C [22,27,45]. Since different indices can contribute to this identification, it is not recommended to rely only on a single one [51], but the two classes above are complementary. Therefore, the scoring applied, based on the resilience and productivity capacity of a genotype, has been previously used in other crops like wheat, beans, and sweet potato, and with other stressors like heat [34,52–55].

The YSSI, which best integrates the scored indices responses, was highly correlated with both yield under full and restricted irrigation and gave a good contrast between genotypes (Table 2). It indicates that Huayro negro, Sumaq sunqu rojo, Yana galla shaco, Viuda, Gara callhuan, Huayro plomo, Shogo chata, Uncuy, and Muru dólar would have good potential yield and tolerance to drought stress, and can be related to Fernandez [27]'s group A. The subsequent ones in this table had contrasting responses between the two classes of indices. Those like Shiri, Orgon runtush, and Chiaquil negro obtained red numbers despite having a better yield under stress-free conditions than many others, due to being seriously affected by water restriction (more than 80% of yield reduction), and could correspond to group B. Finally, at the bottom of Table 2, the landraces with low scores for all the indices like Cera monilla, Huanuco suytu, Merino, and Puka ranra ñahui showed a poor performance in both scenarios and could represent the group D, which would be the least recommendable for breeding purposes.

When the breeder seeks adapted cultivars, selection should be based on tolerance indices calculated from yield under both conditions [30]. However, there are those who suggest that yield under stress conditions is not always the most appropriate selection trait, and it is more recommended to evaluate physiological traits [56]. It should also be considered that within the large set of genotypes evaluated in this trial, different phenological

lengths could influence index responses, for example, an early maturing variety could be considered an escape strategy rather than tolerance or resistant adaptation. Therefore, it is suggested for future studies to divide the population into groups of similar phenology and to perform analyses like this within each group [30].

## 5. Conclusions

The different responses to drought stress within the 79 assessed potato varieties give signs of a broad genetic diversity within the central Andes of Peru. The applied methodology helped to quickly and easily characterize and differentiate the large set of germplasm as a first selection for future breeding evaluations. It was possible to determine those that could have tolerance to drought stress, as well as the least recommended ones for these purposes. Our results provide an informative basis for the characterization of potato diversity in the central Andes of Peru, in pursuit of the conservation of biodiversity and food security in the region. Future studies that include physiological and molecular evaluation of landraces under stress conditions are recommended, emphasizing the genotypes highlighted by this study.

**Author Contributions:** Conceptualization, E.Z., J.I., F.A. and K.M.; methodology, E.Z., K.M., J.I., F.A. and R.P.; software, E.Z., J.I. and C.S.-D.; validation, C.S.-D.; formal analysis, E.Z., F.A., J.I. and C.S.-D.; investigation, I.V., R.P. and D.B.; resources, I.V. and R.P.; data curation, E.Z., K.M., F.A., J.I., R.P. and C.S.-D.; writing—original draft preparation, E.Z., K.M., F.A., R.P., I.V., D.B., G.R., J.I. and C.S.-D.; writing—review and editing, D.B., R.P., G.R. and C.S.-D.; visualization, F.A. and C.S.-D.; supervision, D.B. and G.R.; project administration, G.R. and I.V.; funding acquisition, E.Z. All authors have read and agreed to the published version of the manuscript.

**Funding:** This research was funded by the Central Research Institute of the National University of Pasco Daniel Alcides Carrion through the project RCU N° 0315-2020-UNDAC-C.U. "Identification of Genes Resistant to Biotic and Abiotic Stress, and Physiology of Native Pope Varieties of the Pasco Region".

**Institutional Review Board Statement:** Not applicable.

**Informed Consent Statement:** Not applicable.

**Data Availability Statement:** Data will be made available upon request to the corresponding author.

**Acknowledgments:** The authors thank Eudys Gavilan and Holbein Poma for their technical support in the field.

**Conflicts of Interest:** The authors declare no conflict of interest.

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
