# Peer review of "Screening Potato Landraces to Cope with Climate Change in the Central Andes of Peru"

_2037-0164, doi:10.3390/ijpb14040085_

Round 1
Reviewer 1 Report
Comments and Suggestions for Authors
The Manuscript ID ijpb - 2718882 “Screening potato landraces to cope with climate change in the Central Andes of Peru” presents an interesting and timely topic as most countries are experiencing the effects of climate change. The paper requires major revisions especially in the discussion of the results. The discussion should compare and relate new or major findings in the results to the existing body of knowledge in the field, in terms of improvement or further advance of current knowledge, and overall significance and contribution to the field.
Line 3: It is not clear what social affects affect food security in the region.
Line 10: evidenced-> showed or demonstrated
Lines 11 – 13: The sentence is not clear. Rephrase the sentence for clarity
Lines 56 – 58: What was the %yield loss of 918 potato genotypes due to drought stress? This needs to be captured to show the magnitude of the effect of drought stress on potato.
Lines 60 to 64: The last paragraph of the introduction - Here, just before the Objectives statement, there should be a concise Rationale statement (Why this specific study was needed?). What specific knowledge gap existed? Please state concrete hypothesis to achieve the objective of current study. Then provide the objectives of the study.
Lines 81 – 82: What was the weight of the soil in the pots? Was the weight uniform in all the pots?
Lines 85 – 88: How was the field capacity of the control group determined and how was the pot water content maintained? How was water restriction/stress treatments controlled for natural drought stress?
Lines 93 – 97: The numbers for the citations should be transferred and placed after the equations.
Table 2: Provide a footnote on what the colours in the table indicate/refer to.
Lines 151 – 167: In the discussion section, for subsection 4.1, the first background is more than 20 lines and contains many elements that were already mentioned in the introduction section or should be transferred to the introduction section. It should be reduced and be more concise. Interpret and discuss your results, no need of 20 lines as background.
Lines 176 - 191: for subsection 4.2, the first background is also more than 15 lines (whole paragraph) and contains many elements that were already mentioned in the introduction section. It should be reduced and be more concise. Interpret and discuss your results, no need of 20 lines as background.
Lines 192 – 249: In the discussion, the authors have been emphasizing more on agreement with other studies than on the novelty of their own findings. We need to know the implications of the findings for the management for the management of drought stress. Highlight the novelty of your own findings and implications. The corroboration with previous studies is not that important.
Line 252: variety of-> different
Comments on the Quality of English Language
Minor English language editing is required.
Author Response
Response to comments made by reviewers to the manuscript ijpb-2718882: “Screening potato landraces to cope with climate change in the Central Andes of Peru”
We thank the reviewers for their learnt judgment and very constructive comments that have enhanced the quality of our manuscript. The modifications made to the manuscript in response to specific comments are detailed below:
REVIEWER #1
Comments
The Manuscript ID ijpb - 2718882 “Screening potato landraces to cope with climate change in the Central Andes of Peru” presents an interesting and timely topic as most countries are experiencing the effects of climate change. The paper requires major revisions, especially in the discussion of the results. The discussion should compare and relate new or major findings in the results to the existing body of knowledge in the field, in terms of improvement or further advancement of current knowledge, and overall significance and contribution to the field.
- Line 3: It is not clear what social effects affect food security in the region.
Thanks for the observation. Line 3 in the abstract refers to what is detailed in the Introduction section (Lines 42-50) following Lüttringhaus et al. (2021) who explain that in addition to the weather changes that directly affect agriculture, there are social factors that contribute to the vulnerability of the people in the Andean region. For example, the urbanization of rural areas is reducing cropping areas, the change of diet preferences which implies that new generations prefer popular meals and leaving the traditional ones, and the youth mobilization from rural to urban areas looking for a better education and job opportunities. This last is causing a generational gap because the elders stay in the rural areas and cannot transmit traditional knowledge related to agriculture to young generations.
Lüttringhaus, S., Pradel, W., Suarez, V., Manrique-Carpintero, N. C., Anglin, N. L., Ellis, D., Hareau, G., Jamora, N., Smale, M., Gómez, R. Dynamic guardianship of potato landraces by Andean communities and the genebank of the International Potato Center. CABI Agriculture and Bioscience 2021, 2(1), 45.
- Line 10: evidenced-> showed or demonstrated
Following the suggestion, the term was replaced by one more appropriate according to the context (Line 10).
- Lines 11 – 13: The sentence is not clear. Rephrase the sentence for clarity
For a better understanding, the comment was modified as follows (Lines 12-14):
It is recommended to complement these studies with physiological and molecular evaluations in stress situations, especially in those with tolerance qualities highlighted in this study, and thus promote the conservation of the biodiversity of this region.
- Lines 56 – 58: What was the %yield loss of 918 potato genotypes due to drought stress? This needs to be captured to show the magnitude of the effect of drought stress on potato.
This information was detailed in Cabello et al. (2012) and included in the Introduction section (Lines 59-62):
This has been demonstrated by Cabello et al. (2012), who subjected 918 potato accessions (clustered into improved varieties, genetic stocks, and landraces), to a 62% reduction of water supply, resulting in an average yield reduction of 58%, but landraces were only affected by 38%.
This reference has also been added to the list in our manuscript:
Cabello, R., De Mendiburu, F., Bonierbale, M., Monneveux, P., Roca, W., & Chujoy, E. (2012). Large-Scale Evaluation of Potato Improved Varieties, Genetic Stocks and Landraces for Drought Tolerance. American Journal of Potato Research, 89(5), 400–410. https://doi.org/10.1007/s12230-012-9260-5
- Lines 60 to 64: The last paragraph of the introduction - Here, just before the Objectives statement, there should be a concise Rationale statement (Why this specific study was needed?). What specific knowledge gap existed? Please state concrete hypothesis to achieve the objective of current study. Then provide the objectives of the study.
The particular characteristics of agriculture and climate in the Andes, together with the possible impacts of climate change, put the food security of the people of this region at high risk. Local farmers have inherited knowledge that helps preserve crop diversity (Lines 45-47) and can help adapt to changes not only in this but also in other vulnerable places. Indeed, diversity-based risk mitigation is a realistic way to adapt to environmental change (Agrawal et al. 2021) (Lines 51-52). Such knowledge is not adequately transmitted to future generations for reasons also discussed in the Introduction section (Lines 47-50). Furthermore, there is a gap between traditional knowledge and science, which we want to reduce from the academic side by initiating a characterization of local varieties selected by farmers.
The last paragraph was divided into two where the (new) penultimate one (Lines 42-56) talks about the threats facing agriculture in the Andes, especially for potato cultivation, and the reasons to do this type of study. The last paragraph (Lines 57-71) shows some precedents and our interest in contributing to the limited information available for regions like ours.
We have modified the last lines to reflect better what was stated (Lines 66-71):
However, the scientific information about the conservation of potato landraces by small-scale farmers is scarce [Luttringhaus et al. 2021], evidencing a gap between traditional knowledge and science. This study aimed to reduce this gap by describing the response to drought stress of 79 potato landraces traditionally grown in the central Andes of Peru (Pasco region) based on their yield and tolerance indices previously tested in this and other crops.
This new reference has been added to the list in our manuscript:
Agrawal, T., Hirons, M., & Gathorne-Hardy, A. (2021). Understanding farmers’ cropping decisions and implications for crop diversity conservation: Insights from Central India. Current Research in Environmental Sustainability, 3, 100068. https://doi.org/10.1016/j.crsust.2021.100068
6. Lines 81 – 82: What was the weight of the soil in the pots? Was the weight uniform in all the pots?
The average weight of the soil substrate in each pot was 4.5 kg, and we tried to have a uniform weight in all the pots. The substrate weight was added to the text (Line 90).
7. Lines 85 – 88: How was the field capacity of the control group determined and how was the pot water content maintained? How were water restriction/stress treatments controlled for natural drought stress?
To determine the field capacity of a pot, tests were carried out on 5 pots to which different volumes of water were added progressively until the complete humidity of the substrate was reached. This was achieved with 2 liters of water per pot. To control the water content in the pots, tensiometers (Irrometer ® IR Series) were placed at a depth of 15 cm in some pots. The controls had to maintain a value of 20 cbar. Initially, irrigation was given with an approximate frequency of once a week and then increased to twice a week depending on the developmental stage of the plant. The restriction in stress treatments was applied at the beginning of flowering and for 16 days, where 4 consecutive irrigation pulses were avoided. In this case, the tensiometers reached values of 45 cbar (Lines 91-95).
8. Lines 93 – 97: The numbers for the citations should be transferred and placed after the equations.
The references for each equation were relocated as requested (Page 3).
9. Table 2: Provide a footnote on what the colors in the table indicate/refer to.
Following your suggestion, a footnote explaining the meaning of colors was added to Table 2 (Page 8).
10. Lines 151 – 167: In the discussion section, for subsection 4.1, the first background is more than 20 lines and contains many elements that were already mentioned in the introduction section or should be transferred to the introduction section. It should be reduced and be more concise. Interpret and discuss your results, no need of 20 lines as background.
Subsection 4.1 has been reduced from a length of 24 to 18 lines where not only a brief context is given to understand our yield results but also the possible causes of their values, in addition to their usefulness is discussed (Lines 155-172).
11. Lines 176 - 191: for subsection 4.2, the first background is also more than 15 lines (whole paragraph) and contains many elements that were already mentioned in the introduction section. It should be reduced and be more concise. Interpret and discuss your results, no need of 20 lines as background.
The first paragraph of Subsection 4.2 has been reduced from a length of 16 to 13 lines, where the information given in previous sections was omitted, but we have maintained the details we considered necessary to understand our interpretation of the index responses in the following paragraphs where are widely discussed (Lines 174-186).
12. Lines 192 – 249: In the discussion, the authors have been emphasizing more on agreement with other studies than on the novelty of their own findings. We need to know the implications of the findings for the management for the management of drought stress. Highlight the novelty of your own findings and implications. The corroboration with previous studies is not that important.
In the two coming paragraphs (187-210) we tried to understand the indices responses, corroborating with previous results, so we can trust that what we got was realistic. After choosing the best describers, which means, the one that contains more information we characterize our genotype set according to it.
The section 4.3 shows the characterization we made to the genotypes, giving the highlights we consider as our contributions. The novelty of our work lies in presenting unprecedented information for our study area, evidencing the diversity of landraces with potential tolerance and also high vulnerability to drought stress, known by local farmers but not necessarily by the academia. Having worked with indices widely used in publications on other crops and places, the fact that our results agree with them gives value to our intention to study local biodiversity in a more in-depth analysis, and eventually contribute with solutions to adapt to the coming changes.
13. Line 252: variety of-> different
In accordance with the reviewer’s observation, the term was replaced for a better expression of the idea (Line 247).

Reviewer 2 Report
Comments and Suggestions for Authors
The main goal of this short article is to characterize the response to drought stress of 79 potato landraces traditionally grown in the Central Andes of Peru (Pasco region) as a first selection for future breeding evaluations, applying yield based parameters such as mean productivity (MP), geometric mean productivity(GMP), stress tolerance (STI), stress susceptibility (SSI), and tolerance index (TOL), and a scoring methodology that integrates all of them into a single descriptor. The study has evidenced the wide genetic diversity within the landraces and discerned some of them with high resilience and productivity qualities, as well as some others - highly drought susceptible. The study is a good basis for more detailed physiological and molecular evaluation of selected local landraces under stress conditions.
The article is clearly and concisely written, with good justification in the Introduction of its importance for biodiversity conservation and food security cover in the region The methodological part contains the necessary details, appropriate statistical analysis is applied, the visual presentation of the results is clear and convincing, the discussion is without unnecessary speculations. Generally, the article is leaning on another work published 10 years ago (Cabello, R., Monneveux, P., De Mendiburu, F., Bonierbale, M. Comparison of yield based drought tolerance indices in improved varieties, genetic stocks and landraces of potato (Solanum tuberosum L.). Euphytica 2013, 193(2), 147–156) and in this respect does not claim for originality. Its usefulness is the simplicity and straightforwardness as a reliable screening approach. It will be nice if the authors can present some phenological data (they comment in the discussion “different phenological lengths could influence index responses, for example, an early maturing variety could be considered an escape strategy rather than tolerance or resistant adaptation”). If such data are collected, this will add value to the article. I hope in the near future, in their next article, the focus will be on selected landraces among the differently stress-responding ones, and on molecular parameters underlying the differences in their performance under drought stress.
Author Response
Response to comments made by reviewers to the manuscript ijpb-2718882: “Screening potato landraces to cope with climate change in the Central Andes of Peru”
We thank the reviewers for their learnt judgment and very constructive comments that have enhanced the quality of our manuscript. The modifications made to the manuscript in response to specific comments are detailed below:
The authors appreciate the comments. We discuss the possibility of considering phenological states to have more homogeneous scenarios in which landraces or varieties are compared, and those with truly tolerant qualities are identified. However, this was found in the literature after conducting our experiment and we did not consider this aspect, but it will be for future studies.

Round 2
Reviewer 1 Report
Comments and Suggestions for Authors
The authors have addressed my comments adequately.
Comments on the Quality of English LanguageMinor editing of English language is required.